# Towards Optimized Photoluminescent Copper(I) Phenanthroline-Functionalized Complexes: Control of the Photophysics by Symmetry-Breaking and Spin–Orbit Coupling

**DOI:** 10.3390/ma15155222

**Published:** 2022-07-28

**Authors:** Christophe Gourlaouen, Chantal Daniel

**Affiliations:** Laboratoire de Chimie Quantique Institut de Chimie UMR 7177, CNRS, Université de Strasbourg, 4, Rue Blaise Pascal CS 90032, F-67081 Strasbourg, France; c.daniel@unistra.fr

**Keywords:** luminescence, TADF, copper(I) phenanthroline functionalization, density functional theory, nuclear relaxation, symmetry-breaking spin–orbit coupling

## Abstract

The electronic and structural alterations induced by the functionalization of the 1,10-phenanthroline (phen) ligand in [Cu(I) (phen-R_2_)_2_]^+^ complexes (R=H, CH_3_, *tertio*-butyl, alkyl-linkers) and their consequences on the luminescence properties and thermally activated delay fluorescence (TADF) activity are investigated using the density functional theory (DFT) and its time-dependent (TD) extension. It is shown that highly symmetric molecules with several potentially emissive nearly-degenerate conformers are not promising because of low S_1_/S_0_ oscillator strengths together with limited or no S_1_/T_1_ spin–orbit coupling (SOC). Furthermore, steric hindrance, which prevents the flattening of the complex upon irradiation, is a factor of instability. Alternatively, linking the phenanthroline ligands offers the possibility to block the flattening while maintaining remarkable photophysical properties. We propose here two promising complexes, with appropriate symmetry and enough rigidity to warrant stability in standard solvents. This original study paves the way for the supramolecular design of new emissive devices.

## 1. Introduction

The emissive properties of bis-phenanthroline substituted Cu(I) complexes have been intensively explored starting in the late 70s [1,2,3,4] because of intriguing luminescence properties extremely sensitive to structural effects driven by the surrounding ligands [5,6,7,8,9,10,11,12,13,14,15,16,17,18,19,20,21,22,23,24,25,26]. The development of time-resolved spectroscopies and pump–probe X-ray experiments has allowed the mapping of ultrafast kinetics that precede the formation of potentially long-lived triplet T_1_ excited states [26,27,28,29,30,31,32,33,34,35,36,37,38,39,40,41,42,43]. The decay mechanism involves several intermediate singlet and triplet excited states coupled vibronically and by spin–orbit [44,45,46,47,48,49], the influence of the solvent being significant on the emission properties [50,51]. The relationship between the observed structural distortions, triggered by visible irradiation, and the excited state dynamics is not clear enough for a rigorous rationalization of the luminescent behavior of a variety of more or less constrained complexes proposed over the years. The structural reorganization in the active metal-to-ligand charge transfer (MLCT) state, from the pseudo tetrahedral D_2d_ symmetry ground state to a flattened D_2_ symmetry excited state, has been investigated combining various spectroscopies with the functionalization of the phenanthrolines [31,33,39,40,52,53,54] (thereafter phen). The hope is to inhibit, by steric hindrance, the early pseudo-Jahn–Teller (PJT) distortion induced by the MLCT electronic transition and to prevent the formation of the flattened structure, critical to generate the appropriate long-lived ^3^MLCT state and high photoluminescence quantum yields. However, the synthesis of stable complexes with bulky ligands is still a challenge and the resulting photophysical properties do not always live up to expectations [55,56,57,58].

The present theoretical study is dedicated to a comparative investigation of the structural, optical and emissive properties of five [Cu(I) (phen-R_2_)_2_]^+^ complexes (R=H (**1_H_**), CH_3_ (**1_Met_**), *tertio*-butyl (**1_tBu_**) and alkyl-linker (**2** and **3**), Figure 1) of increasing steric complexity associated with symmetry breaking. The potential energy profiles associated with the low-lying singlet and triplet excited states as well as their spin–orbit coupling (SOC) as a function of the structural deformations are scrutinized in order to rationalize the photophysical behavior within the series and to point to the consequence of symmetry-breaking on emission properties. In contrast to previous theoretical studies, the implication of upper triplet states and advanced spin–orbit effects are considered going beyond an oversimplified S1/T1 two-state model based on frontier molecular orbitals.

## 2. Computational Method

A first set of calculations has been performed with GAUSSIAN 09 (version D01) [59] at a density functional theory (DFT) level of theory with B3LYP functional [60] in CH_2_Cl_2_ through a PCM model [61]. All atoms were described by Pople’s 6–31+G** basis set [62]. Structures in the ground state and in triplet states were fully optimized and the nature of the encountered stationary point was checked by frequency analysis. Minima were characterized by a full set of real frequencies and the transition state by one imaginary frequency. Dispersion corrections were introduced through Grimme’s corrections [63].

A second set of calculations was performed with the ADF 2019 package [64]. The complexes (Figure 1) electronic ground state and low-lying excited singlet and triplet state structures have been optimized by means of DFT using the B3LYP functional [60] and triple-ζ basis sets for all atoms [65]. Scalar relativistic effects were included using zeroth-order relativistic Hamiltonian [66]. Grimme’s corrections have been applied to consider dispersion effects [67]. Calculations have been performed in CH_2_Cl_2_ within the COSMO (conductor-like screening model) approximation [68]. The absorption spectra were computed at the TD-DFT level, including perturbative spin–orbit effects [69,70]. The Tamm–Dancoff approximation (TDA) was employed in order to avoid triplet instability problems [71].

Cuts of the potential energy surfaces along the key nuclear coordinates (Figure 2), namely the Cu–phen distance (Cu–X_1_), the rocking angle (X_1_–Cu–X_2_) and the dihedral flattening angle (C_1_–X_1_–X_2_–C_2_), were obtained with remaining coordinates frozen at the ground state structure with ADF.

The excited state analysis was performed by TheoDORE, a package for theoretical density, orbital relaxation and exciton analysis [72].

## 3. Results

### 3.1. Ground State and Excited State Structural Properties

The structural data, namely a fully optimized electronic ground state and lowest T1 excited state optimized with GAUSSIAN are given in the Appendix A. The complexes are structurally flexible even in the ground state (GS) due to the weak inter-ligand interactions. This is illustrated by the variety of **1_H_** GS crystal structures [73,74,75,76,77,78,79,80] (Appendix A), for which the structural parameters depend on the nature of the counter-anion and of the solvent in which the compounds were crystallized. The Cu–N distances vary from 1.996 Å to 2.081 Å, the X_1_–Cu–X_2_ angle from 144.8° to 180.0° and the C_1_–X_1_–X_2_–C_2_ dihedral from 33.9° to 90° (Appendix A). The situation is similar for **1_Me_** [81,82,83,84,85,86] (Appendix A). In both cases, the computed GS structure is within the range of experimental data. The situation is different for **1_tBu_**, the bulkiness of the tertio-butyl group prevents the significant distortion of the geometry. However, this group introduces a new degree of freedom in the structure according to the orientation of the tertio-butyl groups. For each of them, there are two possibilities, a methyl group can point towards the copper cation or to the opposite of the cation (Appendix A). This generates six possible conformers (Figure 3, Appendix A), four of them (γ, δ, ε, ζ) exhibiting energy differences lower than 3 kcal mol^−1^ and separated by barriers lower than 7 kcal mol^−1^. This implies that in the solution there is a thermal distribution of the conformers in fast equilibrium. This is supported by the experimental structures of **1_tbu_** [87]. Depending on the complex environment, the structure can adopt different conformations in the crystal (Appendix A). According to frequency analysis, the GS structures of **1_H_** and **1_Me_** are of D_2d_ symmetry point group, that of **1_tBu_** most stable conformer (δ) is only C_2_. **2** exhibits C_2v_ symmetry and **3** has C_s_ symmetry (Appendix A).

The structural evolution is complex during excited state relaxation. In the optimized structures of the T_1_ excited state of **1_H_** and **1_Me_**, a significant flattening of the complex is observed leading to a D_2_ symmetry structure. The C_1_–X_1_–X_2_–C_2_ angle falls to 41.5° for **1_H_** and to 65° for **1_Me_** (Appendix A). In contrast, for **1_tBu_** (steric constraints, Appendix A), **2** and **3** (ligand rigidity, Appendix A), there is almost no flattening. The flattening generates two degenerated minima in 1_H_ and 1_Me_, depending on the rotational direction (Scheme S1). These two structures are connected by a transition state (thereafter TS1) of 12.9 kcal mol^−1^ for **1_H_** and 4.4 kcal mol^−1^ for **1_Me_**.

The symmetry of **1_Me_** and **1_tBu_** is also reduced by the Cu–N bond breathing. One phen ligand is associated with a significant shortening of the Cu–N bonds, between 0.05 and 0.20 Å. The relaxation of the second phen ligand depends on the steric hindrance with a Cu–N bond shortening in **1_Me_** (Appendix A) and a lengthening in **1_tBu_** (Appendix A). A similar distortion operates in complexes **2** and **3**. The bond breathing generates for each complex (except for **1_H_**) two minima in T1 connected by a transition state (therafter TS2). These minima are strictly degenerate for **1_Me_** and **1_tBu_**, the two phen being identical in these molecules. In **2** and **3**, the linkers induce constraints on the phen ligands and slightly lift the degeneracy of the two minima. The barrier associated with TS2 is very low, 2.4 kcal mol^−1^ for **1_tBu_**. The flattening induces two minima depending on the phen orientations (left and right) in **1_H_** and **1_Me_**. Consequently, the T1 PES is characterized by four minima in **1_Me_** (breathing and flattening, Appendix A), two minima in **1_H_** (flattening), two minima in **2** and **3** (breathing). The case of **1_tBu_** is more complicated. The orientation of the tBu groups generates six possible conformers in the ground state (Figure 3), number potentially doubled in the T1 excited state due to the bond breathing. Though some conformers (α, β) can be neglected due to their relative instability, the others may play a central role in the luminescence properties. More specifically, the most stable δ conformer (Figure 3) will be used as reference for the **1_tBu_** excited state properties described below.

The calculated values of TS1 and TS2 are in favor of non-negligible dynamical effects. The bond breathing (TS2) is associated with low barriers and the occurrence of true minima is uncertain. Furthermore, the barriers increase with the steric hindrance of the R group (none for **1_H_**, and **1_Me_** < **1_tBu_**). The formal oxidation of the copper tends to shorten the Cu–N bonds. However, the ligand repulsion in T1 counterbalances this shortening leading to a pendulum motion: one ligand approaches the copper while the second one moves away from it. This effect increases with the size of the R group. The movement of interconversion of the phen (TS1) is more costly but possible for 1_Me_ at the time-scale of excited state lifetime.

### 3.2. Low-Lying Singlet and Triplet Excited State Properties

#### 3.2.1. Energetics, Absorption Spectra and Spin–Orbit Coupling

The calculated absorption spectra are depicted in Figure 1 on the basis of ADF optimized structure. The transition energies, absorption wavelengths and oscillator strengths associated with the 25 lowest singlet excited states of complexes **1_H_**, **1_Me_**, **1_tBu_**, **2** and **3** are reported in Table 1 and Table 2, respectively. Complexes **1** absorb in the visible at ~490 nm via the 3rd singlet excited state (S3, f > 10^−1^) and in the UV domain (f > 10^−2^) at ~340 nm (S17) and at ~320 nm (S23). The orientation of the tBu groups does not affect the structure of the absorption spectrum of **1_tBu_**, inducing unsignificant shifts of about 25 nm in the position of the absorption bands (Appendix A).

The absorption properties of complex **2** are very similar with three bands at 493 nm (S3), 350 (S18) and 320 nm (S22). The absorption of complex **3** starts at 465 nm (S3, *f* > 10^−1^), slightly blue-shifted as compared to the other complexes. As illustrated in Figure 2, the low-lying singlet excited states of the five complexes are mainly of metal-to-ligand charge transfer (MLCT) character with minor additions of ligand-centered (LC) and ligand-to-ligand charge transfer (LLCT). These states principally arise from HOMO and HOMO-1 orbitals towards LUMO to LUMO+3 (Appendix A).

From TheoDORE analysis (Figure 3), it appears that the low-lying triplet states are essentially MLCT with increasing LC contributions when moving to higher excitation energies. Including spin–orbit corrections does not modify the absorption spectra because the low-lying triplet states do not gain intensity by SOC, as illustrated by the data reported in Table 3 and Table 4, which describe the transition energies of the “spin–orbit” states together with associated oscillator strengths and state mixing. Table 5 reports spin–orbit mixing between the low-lying singlet and triplet states.

Whereas SOC has no effect on the absorption process, we may expect a large influence of SO effects on the early time photophysics. Indeed, within 0.28 eV, we find three singlet (S3, S2, S1) and four triplet (T4, T3, T2, T1) states, all MLCT and potentially activated by absorption, vibronic and spin–orbit coupling for driving the ultrafast decay observed experimentally [38] and tentatively analyzed by quantum dynamics simulations [45] for complex **1_H_**. These seven states, significantly coupled by SOC, lead to a high density of spin–orbit states for all complexes (Table 3 and Table 4) between 2.24 eV (E1) and 2.53 eV (E15) in favor of an ultrafast S3 to T1 decay via an efficient spin-vibronic mechanism induced at Franck–Condon by Cu–N breathing mode activation within ~100 fs and at a longer time-scale (~400 fs) by the PJT distortion, in the case of D_2d_ complexes **1_H_** and **1_Me_** [44,48,49,50]. In the highly symmetric molecules, singlets S3, S2 and S1 are strongly coupled with triplet T1, T4 and T2, respectively; T4 and T3 being strongly coupled to T1 and T2, respectively (Table 5). Breaking the symmetry in the sterically-hindered complex **1_tBu_**, complexes **2** and **3** do not drastically modify the energetics but influence the SO interactions, potentially increasing the singlet–triplet interactions at Franck–Condon, in particular S1/T1 and S3/T3 (complex **1_tBu_**; complex **2**) or S2/T1, S3/T2, S3/T3 and T2/T1 in complex **3**. However, inhibiting the PJT distortion in these complexes will decrease some of the vibronic interactions. Altogether and without experimental data and/or quantum dynamics simulations for these molecules it is difficult to conclude as to the consequences of the phen substitution on the ultrafast S3 to T1 decay.

From the above considerations we may expect concurrent elementary processes to occur upon irradiation in the MLCT band at about 490 nm. The branching ratio between the different scenario will depend on the experimental conditions and ligand substitutions. We have to distinguish between highly symmetric molecules with small steric constraints (complexes **1_H_**, **1_Me_**), for which flattening nearly inhibits luminescence and sterically-demanding molecules as complex **1_tBu_** is characterized by significant quantum yields (ϕ^em^ > 10^−2^) and rather long lifetimes (a few hundred of ns), challenging the [Ru(bpy)_3_]^2+^ complex. These case studies, although well documented, are not fully rationalized and fail at explaining the differences in observed quantum yields and lifetimes.

Quantum dynamics simulations are too prohibitive to be performed systematically on the molecules described here, so we scrutinized the cuts of the PES associated with the seven low-lying active excited states discussed above as a function of the Cu–X distances (C_2v_ symmetry constraint), the X_1_–Cu–X_2_ rocking angle (C_s_ symmetry constraint) and the C_1_–X_1_–X_2_–C_2_ dihedral angle (D_2_ symmetry constraint) (Figure 2).

#### 3.2.2. Calculated Potential Energy Curves Associated with the Low-Lying Excited States and Their Spin–Orbit Interactions

A spin-vibronic mechanism is controlled by the distortions at Franck–Condon that induce intrastate coupling leading to a shift of the excited state potentials in position and in energy. This generates critical geometries favorable to efficient non-adiabatic transitions induced by vibronic and SO couplings. Let us examine the case study of complex **1_H_** and the cuts of PES depicted in Figure 4 for this D_2d_ molecule. It should be kept in mind that in the D_2d_ structure the HOMO is of e symmetry (doubly degenerate orbital) as well as the LUMO, generating degenerate S1 and T1 states. Upon geometry distortion, the symmetry is reduced and the degeneracy is lifted. This has no effect on the PES curve (Figure 4) but has some influence on the SOC curves (Figure 5). The S1–T1 SOC is nil for the breathing and rocking modes but not for the flattening due to the electronic reorganization in the excited states upon symmetry reduction.

The breathing of the phen ligands clearly induces the stabilization of S1 and T1 with the formation of two minima at Cu–X = 3.87 Å and 4.07 Å, corresponding to the exciton delocalization on one or the other phen (Figure 4a and Figure 5). Several crossings between S1 and T4, T3 and T2 will favorize an ultrafast population of these triple states. Moreover, the near-degeneracy between T2 and T1 at 3.97 Å is promising for an efficient population of T1 (Figure 3). The rocking angle X_1_–Cu–X_2_ deformation does not modify the ordering of the low-lying excited states and the associated PEC are nearly flat (Figure 4b). More interestingly, the dihedral deformation C_1_–X_1_–X_2_–C_2_ associated with the flattening of the molecule drastically destabilizes the T3 and T4 excited states and significantly stabilizes the S1, T2 and T1 states (Figure 4c). The large S1–T2 SOC at Frank–Condon and the increase of T1–T2 SOC as function of Cu–X with two maxima at 3.87 Å and 4.07 Å, combined with small S1–T1 and S1–T2 energy gaps (<0.15 eV) (Figure 6), favor a T2/T1 exchange of population at the early time (<1 ps), as observed experimentally [38]. These ultrafast processes have been rationalized theoretically by means of dynamics simulations [44,50].

It seems clear that only S1, T2 and T1 will be involved at the longer time-scales (>a few ps) that control the luminescence quantum yields and lifetimes. The parameters which govern the different mechanisms and the branching ratio between them, namely S1–S0 fluorescence, T1–S0 phosphorescence, indirect TADF and non-radiative decay, are the oscillator strengths; the S1/T1, S1/T2 and T1/T2 SOC; the S1–T1, S1–T2 and T1–T2 ΔE energy gaps and their evolution as functions of the structural deformations.

A TADF mechanism is excluded for **1_H_** because S1–T1 SOC is null and the flattening induces both a large increase of the S1–T1 energy gap and a significant decrease of the S1–S0 oscillator strength. The only possibility of a back population of S1 would be via the T2 state driven by a small S1–T2 energy gap and a significant S1/T2 SOC at the equilibrium structure. However, the structural deformations, both the distortion and the flattening, do not induce important S1/T1 SOC, reduce S1/T2 SOC and increase the energy gaps (Figure 6).

At the T1 optimized structure, the spin–orbit state E1, E2 and E3 oscillator strengths vary drastically as functions of the key nuclear coordinates (Figure 2), as illustrated in Figure 7. The flattening (Figure 7c) quenches the phosphorescence. Moreover, according to the data reported in Table 6 for **1_H_**, the calculated emission wavelengths associated with S1 → S0 and T1 → S0 are too low to make these transitions radiative.

Let us focus now on the **1_tBu_** complex, in which the flattening is nearly inhibited. The only allowed substantial structural deformation is the breathing, which generates S1 and T1 potentials characterized by two minima, in a manner similar to **1_H_** but totally dissymmetric with one potential well at 3.90 Å, where the system can be trapped either in S1 or in T1 (Figure 8).

In **1_tBu_**, ΔE_S1–T1_ amounts to 0.12 eV on average with very small variations and the symmetry breaking activates S1/T1 SOC. In addition, the structural deformation generates a S1/S0 oscillator strength of ~10^−3^ in **1_tBu_** as compared to ~10^−13^ in **1_H_**. At Franck–Condon, the S1–T1 SOC is large, warranting the efficient population of T1. As soon as T1 is populated the system evolves to the T1 minimum at 3.90 Å, where S1 (Figure 9) can be back populated through S1–T1 SOC, assisted by large T1/T2 and S1/T2 SOC (Figure 8a). Consequently, the system may be the seat of T1 → S0 phosphorescence and of both direct (at Franck–Condon) and TADF (at 3.90 Å) S1 → S0 fluorescence. The contribution of T2 to the phosphorescence cannot be excluded. This explains the unique photophysical properties of **1_tBu_** developed after absorption at 425 nm, namely a long-lived MLCT emission (λ_em_, 599 nm; τ, 3260 ns) and the largest quantum yield (ϕ, 5.6%) of all [Cu(R_2_phen)_2_]^+^ complexes [53]. This mechanism, corroborated by the data reported in Table 6 for **1_tBu_**, namely the deformation, emission and stabilization energies, emission wavelengths, singlet-triplet energy gaps, SOC and oscillator strengths at the excited state optimized structures, explains the occurrence of a superposition of phosphorescence and TADF contributions in the steady-state emission spectra as discussed experimentally for a number of new sterically-hindered complexes [54,88,89,90].

The proposed mechanism for the two case studies reported above, namely **1_H_** and **1_tBu_**, may be drastically modified by the experimental conditions (solvent, temperature, etc.). This points to the weakness of the two-state S1/T1 model based on frontier molecular orbitals defined for one minimum. The presence of six identified conformers in the ground state PES of **1_tBu_** (**α**, **β**, **γ**, **δ**, **ε** and **ζ**, Figure 3), multiplied by two in the S1 and T1 excited states (by breathing motion, Appendix A) thwarts the above mechanism on a long time-scale. Based on the low calculated energy barriers (<7 kcal mol^−1^) the free rotation of the tBu groups in solution and the thermal distribution of the different conformers are ensured (Table 7).

The emission properties (deformation, emission, stabilization energies, emission wavelengths, oscillator strengths, S1–T1 energy splitting and SOC) computed at S1 and T1 minima are reported in Appendix A. The C_1_ symmetry of the T1 state originated from the δ conformer warranties a significant S1–T1 SOC (10.1 cm^−1^) and a small S1–T1 splitting (0.143 eV) favorable to TADF associated with a highly emissive S1 state (f > 10^−4^). In contrast the T1 and S1 states generated by the conformer ζ are of C_2_ symmetry, resulting in the C_2v_ symmetry reduction in the ground state structure. If the S1–T1 splitting (0.140 eV) is unaffected in the ζ conformer, the S1–T1 SOC is inhibited despite this small symmetry breaking (Appendix A), which deactivates the TADF mechanism. Moreover, the low oscillator strength of S1 (f < 10^−7^) does not support fluorescence. Due to its stability, the ζ conformer is undoubtedly present in the solution and contributes to the emission properties either by phosphorescence or by non-radiative decay. Another possibility is that ζ plays the role of a reservoir: by tBu rotation it may evolve to the γ, δ and ε conformers reactivating the SOC and, thus, the TADF mechanism.

The co-existence of several conformers in the solution drastically complicates the mechanism due to the presence of shallow minima in the PES associated with S1, T1 and S0 accessible on a longer time-scale. Moreover, we may expect conformer specific ultrafast decay channels, as observed in some organic chromophores [91].

Whereas the ultrafast population of the low-lying S1, T2 and T1 excited states is well documented, both theoretically and experimentally for **1_H_**, the longer time-scale non-adiabatic dynamics, including spin-vibronic effects and involving these three key states, has to be discovered. The present study paves the way to more sophisticated dynamical simulations in the ns time-scale.

### 3.3. Towards Supramolecular Design

Despite its remarkable photophysical properties, the *tertio*-butyl-substituted complex **1_tBu_** suffers from serious drawbacks, namely instability in various solvents [53]. The steric congestion due to the *tertio*-butyl groups favors the decoordination of one of the substituted phen by a solvent molecule (as CH_3_CN). This can be illustrated by the calculated complexation energies of the second phen onto the [Cu(phen)]^+^ complex to give [Cu(phen)_2_]^+^. The ΔG of complexation amounts to −31.6 kcal mol^−1^ for **1_H_** and −38.5 kcal mol^−1^ for **1_Me_** but decreases to −19.8 kcal mol^−1^ for **1_tBu_** (computed with GAUSSIAN, in CH_3_CN). In light of the performance of **1_tBu_**, other functionalizations of the phen have been explored in position 2 and 9 of the phen to prevent flattening by steric congestion. However, none of the tested complexes exhibit performances similar to **1_tBu_**. For instance, inter-ligand interactions may cause flattening even in the singlet ground state, as with 2,9-diphenylphenantroline [92,93]. To the best or our knowledge, no example of flattening blocked due to the linkage of the two ligands exists, while partial linkage has been explored [56,94,95]. We present here two promising structures illustrating this possibility. They derive from the structure ζ of **1_tBu_**, a link being created between the alkyl groups of position 2 of the two phens. The same link is created with the groups in position 9 (Figure 1). We propose complex **2** (Figure 1), in which a C_2v_ symmetry is retained, and complex **3**, which is asymmetric due to the presence of an isopropyl group.

#### 3.3.1. Complex 2

The linkage constrains the geometry and make the two phen core inequivalent, due to the different orientations of the methyl groups in the linker. Two methyl point towards the copper center and two methyl point outside the complex (Figure 10). The Cu–N distances are significantly larger than in **1_H_** and **1_Me_**, being close to those in **1_tBu_** (Appendix A). The X_1_–Cu–X_2_ and C_1_–X_1_–X_2_–C_2_ angles fit the ideal values, namely 180° and 90°, respectively. The absorption spectrum of **2** is very close to that of **1_tBu_** (Figure 1) with similar transitions (Figure 2 and Figure 3).

Upon excitation, the structural rigidity imposed by the linker leads to the retention of the C_2v_ symmetry. As for the previous complexes, the lowest excited states are mainly MLCT, and the electron transferred to the ligand is localized on one of the phen and generates two almost degenerate minima on the lowest S1 and T1 PESs (Appendix A). All the structures belong to the A2 symmetry point group and their photophysical characteristics are presented in Table 8. They are very close to those of the ζ form of **1_tBu_** with a small singlet–triplet gap (ΔE_ST_ = 0.135 eV) at T1 geometry. However, due to the A2 symmetry of T1 and S1, the SOC S1–T1 is strictly zero, deactivating the TADF mechanism. Furthermore, the oscillator strength associated with S1 is also zero. This first structure proves that the introduction of linkers between the two phen ligands induce structural rigidity preventing any flattening. This results in a small ΔE_ST_ similar to those computed with **1_tBu_**. However, the C_2v_ symmetry of the excited singlet and triplet state disfavors the TADF mechanism by cancelling the SOC between S1 and T1 and the oscillator strength of S1.

#### 3.3.2. Complex 3

The structure of cage **2** is of C_2v_ symmetry, and similarly to the ζ conformer of **1_tBu_**, this disfavors emission. The emission properties of **1_tBu_** are due to its asymmetric conformers. We modified the structure of **2** by deleting three of the methyl groups of the linker and replacing the fourth one with an isopropyl (Figure 1) to introduce an asymmetry in complex **3**. The two phen ligands are no longer equivalents (Appendix A) and the Cu–N bonds especially are significantly different (2.020 to 2.299 Å) in **3** as compared to **2** (2.140 to 2.156 Å). Furthermore, the phen ligand facing the isopropyl moiety is no longer planar. The absorption spectrum of **3** (Figure 1) is similar in shape to that of **2** but blue-shifted by roughly 50 nm. The nature of the singlet transitions (Figure 2) is the same as in the other complexes, being almost exclusively dominated by MLCTs.

We optimized the lowest excited singlet and triplet states of **3** (Table 8, Appendix A, Figure 11). We retrieved the two minima on the S1 and T1 PES, due to the localization of the exciton on each of the phen. The presence of the isopropyl group breaks the symmetry, and the two minima are no longer degenerate. Furthermore, both S1 and T1 minima do not have any symmetry. The consequence is an activation of the SOC between S1–T1 (7.7 cm^−1^). The linkers, by preventing any flattening of T1 structure (Appendix A), retain a small ΔE_ST_ (0.146 eV, Table 8). These values are comparable to that of **1_tBu_** for forms γ, δ or ε (Appendix A), which are those contributing to the emission properties of **1_tBu_**.

## 4. Discussion

The emission properties of the [Cu(phen)_2_]^+^ class of complexes is governed by the TADF mechanism, namely a back-population of S1 from T1. This process depends on two critical parameters, the spin–orbit coupling and the energy splitting between S1 and T1 in the simplest case. Upon excitation, the lowest excited state of these complexes is a MLCT from the copper cation towards the ligand, which induces nuclear relaxation. Two main motions (Scheme S1) are involved. First, the formal electronic state of the copper is Cu^2+^, with only nine electrons in the 3d shell, this leads to the rotation of the phen ligand to form a planar complex, with a half-filled 3dx^2^-y^2^ orbital (the “flattening”). This is the first motion. The second one is due to the localization of the excited electron on the ligand. In the Franck–Condon geometry, the electron is generally delocalized on the two phen. However, upon relaxation, there is a breathing of the Cu–N bonds due to the change of Cu formal oxidation state associated with the localization of the excited electron on only one of the phen (“the breathing”). This potentially generates two minima, as the excited electron could be on one or the other phen. The minima are degenerate if the two phen are identical, and if they are asymmetric, degeneracy is lifted. In addition to these two motions, a last point has to be considered. Indeed, with complex ligands, the conformational flexibility has to be taken into account. **1_tBu_** is a good example with several conformers generated by the relative orientation of the tertio-buthyl groups, each conformer exhibiting its own emission properties. The experimental values are the results of the contribution of the conformers thermal distribution.

The flattening is associated with an increase in the ΔE_ST,_ which disfavors TADF and paves the way to phosphorescence or non-radiative decay. We also observe a decrease in the S1–T1 SOC. The changes associated with the breathing are less important, the variations of the ΔE_ST_ are negligible but the SOC decreases with the distortion of the structure. The prevention of the flattening and limitation of the breathing are mandatory to retain good emission properties. This is achieved with **1t_Bu_**, which bulky *tertio*-butyl groups makes flattening impossible and is associated with the best emission performances. Less bulky CH_3_ groups in **1_Me_** lead to significant flattening with poor emission and **1_H_** is not emissive. However, the bulkiness of the *tertio*-butyl groups is the source of inter-ligand repulsion, destabilizing the complex. In a coordinative solvent, such as CH_3_CN, one of the ligands may be substituted by solvent molecule. To overcome this problem, many complexes have been synthesized with less bulky substituents. However, if the complex stability increases, the less sterically hindered complex leads to significant flattening, weakening the emission performance. The reason for this flattening is that the substituent can adapt its position, as observed with isopropyl [96]. For the latter, the isopropyl rotate and allows a significant flattening. An extreme case can be seen in phenyl rings instead of *tertio*-butyl, in this complex the structure is flattened even in the ground state [92,93]. All the proposed structural modifications rely on the geometry constraints due to substituent repulsion to maintain a geometry close to Franck–Condon geometry in the excited state.

An alternative path is possible by linking the phen ligand together. Some structures have been synthesized but not with linkers between the phens on the same complex. Our detailed theoretical study led to promising molecules with emission characteristics tailored by symmetry breaking and spin–orbit coupling. Due to the structural rigidity, the complex should be stable in a standard solvent, with the departure of one of the phen requiring the departure of both of them. The flattening in excited states is prevented here by linkers instead of steric repulsion. Consequently, the S1–T1 energy gap remains small in complexes **2** and **3** and compares to the calculated value in **1_tBu_**. However, as shown in **1_tBu_**, structures that are too symmetric and close to C_2V_, are not in favor of efficient emission, as illustrated by complex **2** or by some conformers of **1_tBu_**. Complex **3**, with its asymmetric linkers, exhibits the most promising emission characteristics.

## 5. Conclusions

We report here a complete computational study of the structural, optical and photophysical properties of copper(I) phenanthroline-functionalized complexes, including solvent and spin–orbit coupling effects. The study focusses on the parameters which govern the different decay mechanisms and the branching ratio between them, namely S1–S0 fluorescence, T1–S0 phosphorescence, indirect TADF and non-radiative process. For this purpose, the oscillator strengths, the S1/T1, S1/T2 and T1/T2 spin–orbit interactions as well the S1–T1, S1–T2 and T1–T2 energy gaps are calculated as a function of the structural deformations of the complexes. More specifically, three modes are important: the Cu–X breathing, the X–Cu–X angular distortion (X being the center of the functionalized phen ligands) and the flattening mode. A TADF mechanism is excluded for [Cu(I)(phen-R_2_)_2_]^+^ (R=H, methyl) because of a null S1–T1 SOC, a large S1–T1 energy gap and a significant decrease in S1–S0 oscillator strength induced by flattening. Radiative processes are hindered as well. The inhibition of the flattening in [Cu(I)(phen-R_2_)_2_]^+^ (R = *tertio*-butyl) and the identification of six conformers in its electronic ground state, generating twelve local minima in the S1 and T1 potential energy surfaces along the breathing mode lead to a complex mechanism. At the early time, a large S1–T1 SOC activated by symmetry breaking, a small S1–T1 energy gap and a large S1/S0 oscillator strength spawned by structural deformation are in favor of both direct fluorescence (at Franck–Condon) and TADF. An efficient phosphorescence is expected from T1 with a potential contribution of T2. At longer time-scales the co-existence of several conformers in solution drastically complicates the mechanism. These features explain the unique photophysical properties of *tertio*-butyl complex as well as the occurrence of the superposition of phosphorescence and TADF contributions in newly synthesized sterically hindered complexes.

In order to overcome the drawback of instability in solution, the consequence of the decoordination of one phen by a solvent molecule, two complexes with promising photophysical properties are proposed. The new structures are derived from the C_2v_ conformer ζ of the *tertio*-butyl substituted Cu(I) complex by linking the two phen in positions 2 and 9 (complex **2**). The introduction of asymmetry (complex **3**) by appropriate linkers leads to the most interesting molecular cage in terms of photophysical characteristics paving the way for a new supramolecular design.

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
