# Peer review of "Towards Optimized Photoluminescent Copper(I) Phenanthroline-Functionalized Complexes: Control of the Photophysics by Symmetry-Breaking and Spin–Orbit Coupling"

_materials, 2022, doi:10.3390/ma15155222_

Round 1

Reviewer 1 Report

Please, find the Attachment.

Author Response

The bis-phenanthroline substituted Cu(I) complexes reveal the luminescence sensitive to structural effects. However, relationship between the triggered structural distortions and the excited state dynamics is not clear enough for a rigorous rationalization of the luminescent behavior of a variety of constrained complexes proposed. In contrast to previous theoretical studies, the present one goes beyond oversimplified two-state models.

Namely, within the TD DFT (Time-Dependent Density-Functional-Theory) including perturbative spin-orbit coupling effects a computational study of structural, optical and photo-physical properties of five copper(I) phenanthroline-functionalized complexes is reported. The study focuses on the parameters (oscillator strengths, spin-orbit interactions, energy gaps, etc. calculated as functions of the complexes structural deformations) which govern the decay mechanisms and the branching ratio between them, in particular, for luminescence and TADF (Thermally-Activated-Delay-Fluorescence).

To overcome the detected drawback of possible instability in solution (phenanthroline decoordination by the solvent), two complexes with appropriate symmetry and enough rigidity in standard solvents are proposed. And introduction of asymmetry complex by appropriate linkers leads to the interesting molecular cage in terms of paving the way to supramolecular design for new emissive devices.

Several inaccuracies related to numbering should be pointed out:

 - In the text, there are mentioned “Table S1”, ... , “Table S5”, while the tables entitled as “Table 1”, ... , “Table 5” are inserted.

 - In the text, the “Table S4” appears before the “Table S3”.

 - In the text, there are mentioned “Table S9”, “Table S11”, “Table S12” and “Table S13”, while these the tables entitled similarly are not inserted.

 - In the text, the “Table S9” appears after the “Table S11” and “Table S13”.

 - In the text, the “Table S12” appears after the “Table S13”.

 - There are inserted two different tables entitled similarly as “Table 7”.

 - In the text, there are mentioned “Figure S1” and “Figure S2”, while the figures entitled as “Figure 1” and “Figure 2” are inserted.

Paper can be published after the minor revisions are done.

Answer

We thank the reviewer for his/her review. We apologize for the misunderstanding and precise in the main text that these table and figures refer to electronic supporting information. We corrected the order of appearance of Table S3 and S4. Obviously, we made an error during the submission process and the supporting information were missing.

Reviewer 2 Report

Recommendation: Publish after minor revision.

Comments:

The authors presented an in-depth theoretical study of a class of Cu-phen complexes as a promising class of TADF material. The manuscript is well-organized and highly informative, and the reviewer would be happy to recommend it to publish it in Materials with addressing just a few comments.

1.     Unfortunately, I did not see the file of supporting information, but as the information provided by the author is sufficient to provide a stand-alone manuscript, I’m still considering the manuscript well-informative. But if possible, please provide the supporting information file if considering its publication online.

2.     As the authors used Bold font in Tables 1 and 2 to emphasize the states with significant oscillator strength. Most excited states straight from the calculation, especially considering significant steric hindrance, are degenerated and meaningless, experimentally. Would it be a better idea to use bold or another color of font in Figures 2 and 3 to emphasize the characters of the states of interest?

3.     There’re a few studies that provided the transition density of the states of interest to study the excited-state electronic structures. Is it possible to extract such information from TD-DFT data? Including such information provides another prospect of view towards the nature of electronic transitions, especially S1ßàS0, T1ßàS0, and T1ßàS1 transitions, which are very important for TADF materials.

4.     Is it possible to add the FMO picture for the states of interest somewhere in the manuscript or SI? It would be very helpful for experimental chemists/material scientists to further investigate the complex.

Author Response

The authors presented an in-depth theoretical study of a class of Cu-phen complexes as a promising class of TADF material. The manuscript is well-organized and highly informative, and the reviewer would be happy to recommend it to publish it

in Materials with addressing just a few comments.

  1. Unfortunately, I did not see the file of supporting information, but as the information provided by the author is sufficient to provide a stand-alone manuscript, I’m still considering the manuscript well-informative. But if possible, please provide the supporting information file if considering its publication online.

Answer

We are sorry for the missing of the supporting information and we will check their presence during the resubmission process.

  1. As the authors used Bold font in Tables 1 and 2 to emphasize the states with significant oscillator strength. Most excited states straight from the calculation, especially considering significant steric hindrance, are degenerated and meaningless, experimentally. Would it be a better idea to use bold or another color of font in Figures 2 and 3 to emphasize the characters of the states of interest?

Answer

Thank you for the suggestion. We have put in bold the main absorbing state in Figure 2. We did not change the font in Figure 3, for the triplet states, as they do not contribute to the absorption.

  1. There’re a few studies that provided the transition density of the states of interest to study the excited-state electronic structures. Is it possible to extract such information from TD-DFT data? Including such information provides another prospect of view towards the nature of electronic transitions, especially S1→S0, T1→S0, and T1→S1 transitions, which are very important for TADF materials.

Answer

We have added the electron density differences for the S1S0 transitions for 1H, 1tBu and 3 in the main text to illustrate the nature of the state. We did neither add those of the corresponding triplets nor those of 1Me and 2, the transition being exactly of the same nature.

  1. Is it possible to add the FMO picture for the states of interest somewhere in the manuscript or SI? It would be very helpful for experimental chemists/material scientists to further investigate the complex.

Answer

We have added Figure S3 and S4 in the supporting information to illustrate the nature of the frontier orbitals for complex 1Hand 3. We added a sentence in the main text to emphasize that the transitions are generated by the showed orbitals.

Reviewer 3 Report

I reviewed the whole manuscript; the introduction, results, discussion, and conclusions are significance and shown a potential impact for the journal and the readers in the theme. Because the authors report very interesting results on materials with luminescence properties and thermally activated delay fluorescence (TADF). 

Author Response

We thank the reviewer for his/her review.